# SHORT REPORT

# CTCF maintains centromere function and mitotic fidelity

Erin Walsh[1], Thomas Laskarzewski[1,2], Thomas J. Maresca[1,2] and Andrew D. Stephens[1,2,*]

## ABSTRACT

In mitosis, the duplicated genome is aligned and accurately segregated between daughter nuclei. CTCF is a chromatin looping protein that localizes to the centromere in mitosis with an unknown role. We previously published data showing that CTCF constitutive knockdown causes mitotic failure, but the mechanism remained unknown. To determine the role of CTCF in mitosis, here, we used a CRISPR CTCF auxin inducible degron cell line for rapid degradation. CTCF degradation for 3 days resulted in increased failure of mitosis and decreased circularity in post-mitotic nuclei. Upon CTCF degradation, CENP-E was still recruited to the kinetochore and there was a low incidence of the polar chromosomes that occur upon CENP-E inhibition. However, immunofluorescence imaging of mitotic spindles revealed that CTCF degradation caused increased intercentromere distances and a wider and more disorganized metaphase plate, representing a disruption of key functions of the centromere. These results are similar to what is seen upon partial loss of cohesin, an established component of the centromere. Thus, we reveal that CTCF is a key maintenance factor of centromere function, successful mitosis, and post-mitotic nuclear shape.

KEY WORDS: CTCF, Mitosis, Nuclear shape, Centromere, CENP-E, Cohesin

## INTRODUCTION

Mitosis is the stage of the cell cycle where duplicated tethered chromosomes are aligned and then segregated to opposite daughter cells. The mitotic spindle is made of microtubules emanating from opposite sides that must bind via the kinetochore to the centromere of sister chromatids, one centromere bound to each side, to cause biorientation. In metaphase, the centromere acts as a chromatin spring to properly resist microtubule spindle biorientation pulling forces to aid tension sensing and metaphase plate alignment (Lawrimore and Bloom, 2022; Andrade Ruiz et al., 2024; Biggs et al., 2025). This alignment and tension are essential for accurate chromosome segregation in anaphase equally splitting the genome between the two daughter cells (Fonseca et al., 2019; Bunning and Gupta, 2023). Inaccurate chromosome segregation in mitosis can result in aneuploidy and cellular dysfunction that can promote human diseases like cancer (Levine and Holland, 2018). However, many proteins involved in this process have yet to be determined.

[1]Biology Department, University of Massachusetts Amherst, Amherst, MA 01003, USA. [2]Molecular and Cellular Biology, University of Massachusetts Amherst, Amherst, MA 01003, USA.

*Author for correspondence (andrew.stephens@umass.edu)

T.J.M., 0000-0003-2214-8674; A.D.S., 0000-0001-5474-7845

CTCF is a zinc finger chromatin binding protein with a known role in interphase but an unknown role in mitosis. In interphase, CTCF controls chromatin looping to regulate transcription (Hansen et al., 2017; Dehingia et al., 2022). CTCF and cohesin interact to ensure proper looping of the chromatin (Gabriele et al., 2022) and the 3D structure of the chromosome (Samejima et al., 2025). Initial studies have determined that CTCF has a role in mitosis (Wan et al., 2008; Xiao et al., 2015; Funk et al., 2022; Chiu et al., 2023; Watanabe et al., 2023). CTCF has been reported to localize to the centromere (Rubio et al., 2008; Xiao et al., 2015; Del Rosario et al., 2019; Miyata et al., 2021), which is also a hotspot of chromatin looping by condensin and cohesin that aids sister chromatid biorientation and tension sensing in metaphase (Lawrimore and Bloom, 2022). The exact role of CTCF in mitosis remains unclear.

Two core hypotheses remain untested for the mechanism underlying the role of CTCF in mitosis. One hypothesis is that CTCF is essential for recruiting CENP-E (Xiao et al., 2015) a kinetochore-associated kinesin that aids chromosome congression (Yao et al., 1997; Kapoor et al., 2006; Barisic et al., 2015). An alternative hypothesis is that CTCF, a known protein that aids chromatin looping, could aid maintenance of the centromere chromatin loop structure (Lawrimore and Bloom, 2019). Similar to its role in interphase, in mitosis, CTCF might be interacting with cohesin at the centromere, where it is enriched (Kolbin et al., 2025). Destabilization of the centromere causes loss of tension sensing, increased intercentromere distance (Maresca and Salmon, 2009; Chiang et al., 2010) and abnormal separation in anaphase leading to lagging chromosomes or anaphase bridges (Carvalhal et al., 2018). To determine the role of CTCF in mitosis, we used a CTCF-mAID-Clover CRISPR cell line for rapid degradation (Yesbolatova et al., 2020).

## RESULTS AND DISCUSSION

### Multiple day degradation of CTCF increases mitotic errors

To determine whether rapid loss of CTCF disrupts mitotic fidelity, we used the previously generated CTCF-mAID-Clover HCT116 human cell line (Yesbolatova et al., 2020). Cells were treated with 5-Ph-IAA to induce >80% degradation of CTCF-mAID-Clover within hours (Fig. 1A,B; Fig. S1A). CTCF degradation remained constant over 3 days of 5-Ph-IAA treatment as measured by antibody immunofluorescence and western blotting (Fig. 1C; Fig. S1B, Fig. S2). Time lapse imaging of SPY650-DNA every 10 min for 16 h revealed that untreated CTCF-mAID-Clover cells showed low levels of mitotic failure at 2.6±0.8% (mean±s.e.m.) (Fig. 1D). Mitotic errors increased slightly but insignificantly over 1 or 2 days of 5-Ph-IAA treatment to degrade CTCF (Fig. 1D). CTCF degradation for 3 days resulted in a statistically significant increase in mitotic failure to 13.0±1.2% (Fig. 1D). A 3-day 5-Ph-IAA treatment of the parent cell line, lacking the CRISPR modification of CTCF, did not change mitotic failure rates (untreated 1.6±0.4% versus parent 5-Ph-IAA 0.5±0.5% mitotic failure, Fig. S1C). Thus, auxin-induced degradation of CTCF results in a significant increase in mitotic failures after 3 days.

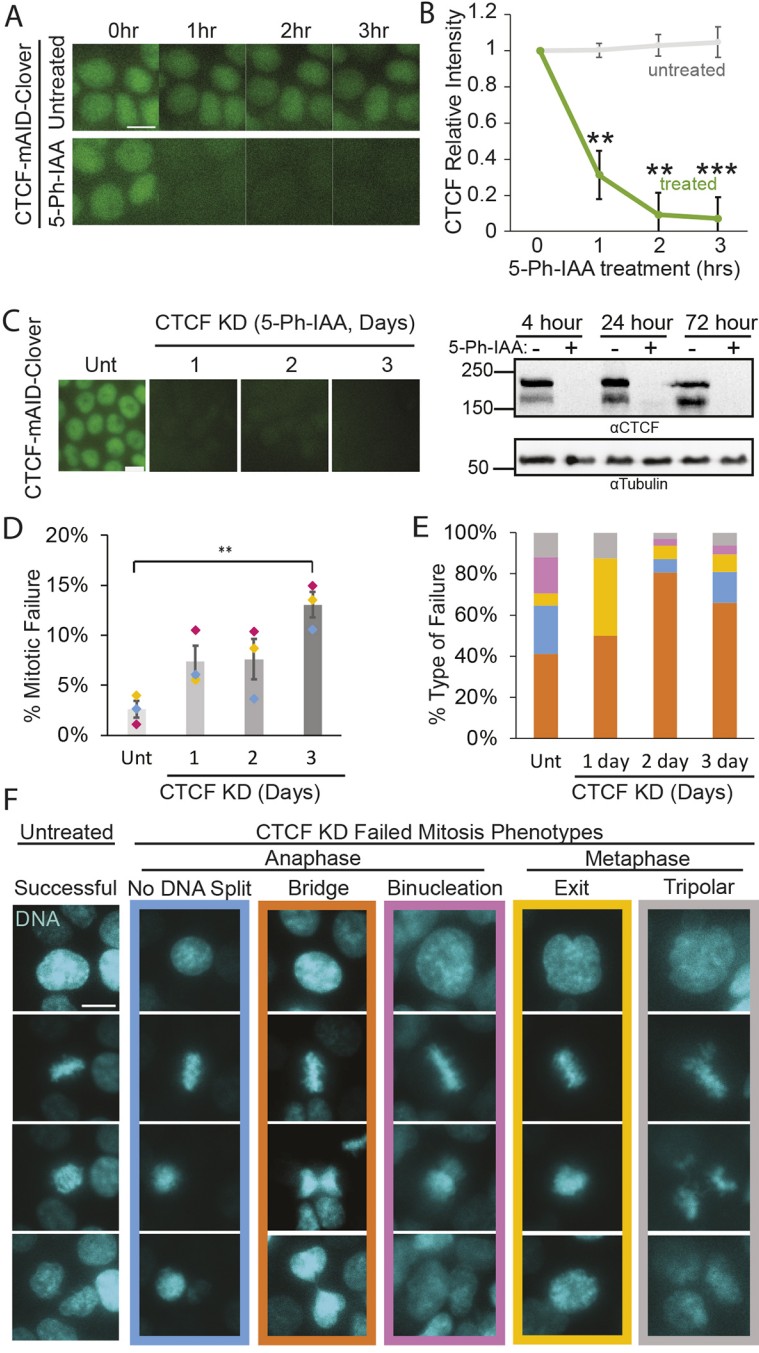

**Fig. 1. CTCF-mAID-Clover degradation causes increased mitotic failure rates mainly through anaphase failure.** (A,B) Example images (A) and graph (B) of CTCF-mAID-Clover relative fluorescence intensity at 0, 1, 2 and 3 h in untreated (Unt) and 5-Ph-IAA-treated cells. *n*>70 nuclei for each of three replicates. Western blots verifying results for 4 h of 5-Ph-IAA treatment in≥85% CTCF knockdown are shown in Fig. S1B. (C) Fluorescence image examples and western blot examples of CTCF degradation after 5-Ph-IAA treatment over 4 h, 1 day and 4 days quantified to be 96.5%, 95.7% and 94.8% knockdown, respectively. Images representative of one biological repeat. (D,E) Graphs of (D) the rate of mitotic failure and (E) the percentage of each type of mitotic failure in untreated, and after a 1-, 2-, and 3-day treatment with 5-Ph-IAA for CTCF knockdown (CTCF KD). Data on mitotic failures come from biological triplicates where each replicate *n*>75 mitoses. Mitotic failure phenotypes include no anaphase separation of the DNA (blue), anaphase bridge (orange), binucleation (purple), mitotic exit (yellow), and tripolar spindle (gray). (F) Example time lapse images of the five phenotypes of mitotic failure imaged with SPY650-DNA. Results in B and D are mean±s.e.m. **$P<0.01$; ***$P<0.001$ [unpaired two-tailed Student's *t*-test (B,E); one-way ANOVA and post hoc Tukey tests (D)]. Scale bars: 10 μm.

Mitotic failures were classified in groups dependent on the defining phenotype with the majority showing anaphase failures (Fig. 1E). The most prominent anaphase phenotype across conditions was an anaphase bridge between the daughter nuclei (orange, Fig. 1E). Less frequently, anaphase failed to split the DNA, resulting in all chromosomes in one daughter nucleus (blue, Fig. 1E) or binucleation in which daughter nuclei separated but remained in one cell (magenta, Fig. 1E). Other minor CTCF knockdown (KD) phenotypes found were metaphase-based tripolar spindles and mitotic exit in which metaphase ended without an attempt to separate the chromosomes or divide into two nuclei (gray and yellow, respectively, Fig. 1E). Overall, the distribution of types of mitotic failure did not change drastically. Furthermore, mitotic failures on average showed a delayed mitosis length relative to successful mitosis across conditions

(Fig. S1D). Thus, the dominant failure upon CTCF degradation remained anaphase separation failures, largely consisting of anaphase bridges.

The absence of effects upon rapid CTCF degradation suggests that it does not establish a key component of the mitotic spindle but instead has a maintenance role that requires more than a single cell division to present. Two days of CTCF degradation in embryonic stem cells are required to affect genome organization measurements (Nora et al., 2017). Thus, our data on CTCF KD shows a similar timeframe for onset of mitotic effects. Overall, CTCF KD disrupts mitotic fidelity through anaphase failures, in agreement with previous work (Xiao et al., 2015; Chiu et al., 2023; Watanabe et al., 2023).

Anaphase bridges can lead to systemic genomic instability (Rodriguez-Muñoz et al., 2022). Concatenations that occur during

DNA replication are usually decatenated by topoisomerase II (Topo II), aided by cohesin, condensin and mitotic spindle microtubule tension, which resolves sister chromatids (Finardi et al., 2020). Thus, CTCF could aid Topo II, cohesin and condensin in the centromere spring during mitosis to resolve sister chromatids under tension, as loss of tension leads to anaphase bridges (Uchida et al., 2021). Partial loss of cohesin during mitosis, an essential component of the centromere chromatin spring, has been shown to produce anaphase bridges, lagging chromosomes and extended mitosis length (Carvalhal et al., 2018). Our data on mitotic failure and phenotype upon loss of CTCF support a possible role in sister chromatid decatenation and/or aiding cohesin in the centromere structure to resist tension.

### Long-term degradation of CTCF leads to abnormally shaped nuclei from post-mitotic nuclear reformation

We reasoned that mitotic failures or even slight disruptions in mitosis could manifest during the post-mitosis nuclear reformation as an abnormal nuclear shape, a hallmark of human disease that can cause dysfunction (Stephens et al., 2019; Kalukula et al., 2022). Thus, the nuclear circularity was determined as a measure of shape, where 1.0 is a perfect circle with all concave edges, with convex edges decreasing this value. To provide a general measure of nuclear shape pre-mitosis, we measured nuclear circularity in the whole population of nuclei in the first time frame of our time lapses. CTCF KD for 2 or 3 days gave a whole-population nuclear circularity, percentage<0.85 circularity and size that did not change significantly from untreated (Fig. 2A,B; Fig. S1E). Next, we measured post-mitosis nuclear reformation shape by determining the nuclear circularity at 1.5 h after anaphase onset. Post-mitosis nuclear circularity significantly decreased upon 3-day CTCF KD (Fig. 2C,D), measured during time lapse imaging of CTCF KD between 72 and 88 h. Using the threshold of abnormal nuclear shape<0.85 circularity, we show a drastic increase in the abnormally shaped nuclei post-mitosis nuclear reformation from untreated at 4.5±3.0% to 31.9±2.6% (mean±s.e.m.) for the 3-day CTCF KD (Fig. 2E). In agreement with decreased post-mitotic nuclear circularity (more abnormally shaped nuclei) accumulating between 3 and 4 days CTCF KD, the whole-population nuclear circularity significantly decreased for the 4-day CTCF KD (Fig. S1F). Thus, taken together CTCF loss disrupts both mitosis and post-mitosis nuclear shape.

Mitotic failures are well known to result in abnormal nuclear shape (Gisselsson et al., 2001). Thus, the role of CTCF in mitosis has downstream implications for the cell in interphase when the miotic chromosome decondense and reforms into the nucleus. Abnormal nuclear shape has been shown to result from disruption of chromosome alignment in metaphase that persists into anaphase and

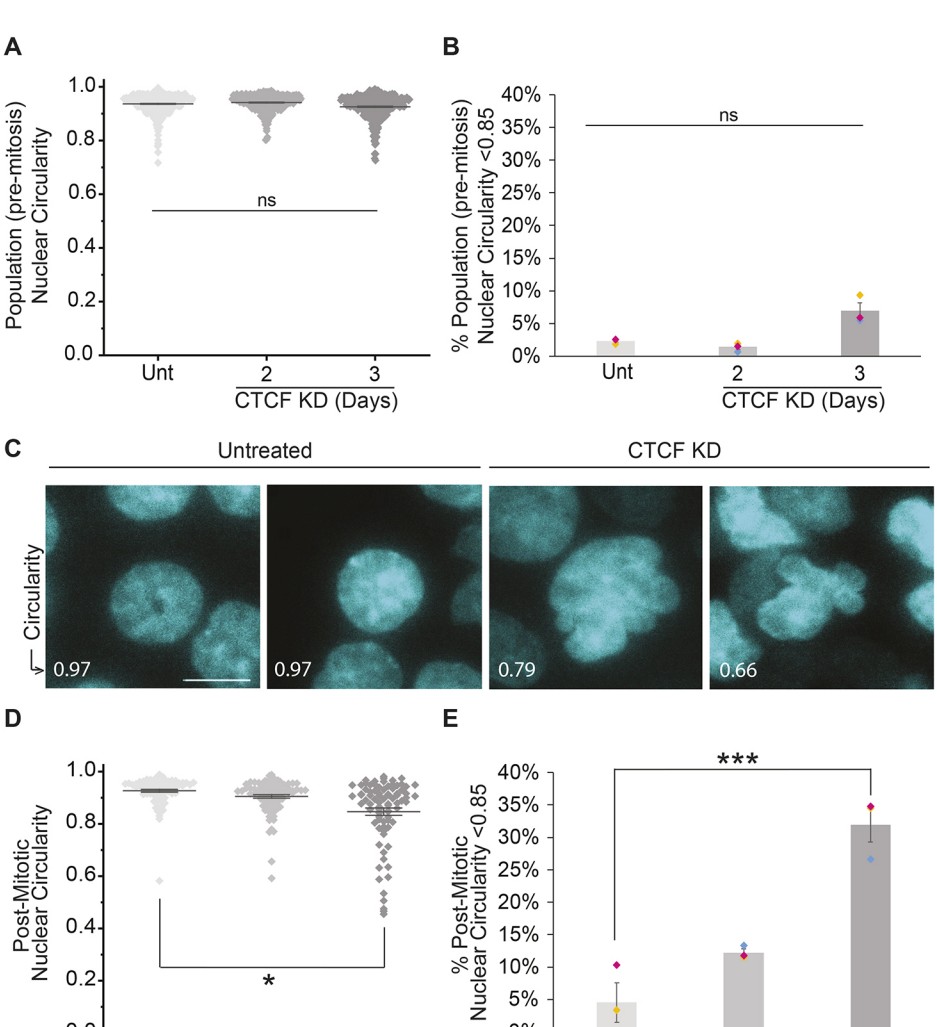

**Fig. 2. CTCF knockdown after 3 days causes decreased nuclear circularity post-mitosis.** (A,B) Graphs of (A) whole population nuclear circularity measurements and (B) the percentage of nuclei with circularity <0.85, representing an abnormal shape, for untreated and 2- or 3-day CTCF KD. These measurements were taken in the first frame of the time lapse to provide pre-mitosis nuclear circularity. Total nuclei population for three replicates each with *n*>350. (C) Example images of nuclei labeled with SPY-DNA (cyan) with the circularity measurement noted in corner. Scale bar: 10 μm. (D,E) Graphs of (D) post-mitotic circularity measurements and (E) percentage nuclear circularity <0.85 for 1.5 h after anaphase onset of untreated (Unt) and 2 or 3 days CTCF knockdown. Three replicates each with *n*>16 from the 16-h time lapse. Results in A, B, D and E are mean ±s.e.m. *P<0.05; ***P<0.001; ns, not significant [Kruskal–Wallis and Dunns tests (A,B); one-way ANOVAs followed by post hoc Tukey tests (D,E)].

telophase (Fonseca et al., 2019). Abnormal nuclear morphology is a hallmark of human disease. Recent findings have revealed that abnormal nuclear shapes that occur because of nuclear blebs in interphase are not only a hallmark but can be a driver of disease progression and overall cellular dysfunction (Stephens et al., 2019; Srivastava et al., 2021; Pho et al., 2023; Chu et al., 2025; Eskndir et al., 2025). Thus, abnormal shape after mitosis could cause similar dysfunctions. Finally, it is well known that CTCF disrupts chromosome organization measures as assessed by Hi-C (Hansen et al., 2017; Pugacheva et al., 2020; Gabriele et al., 2022; Banigan et al., 2023). It is possible that perturbed nuclear shape upon reformation is a key disruptor of chromosome organization. Overall, the role of CTCF in accurate mitotic segregation might extend into the interphase nucleus to determine function.

### CTCF is not required for CENP-E recruitment or function

One hypothesis for the mechanism behind the mitotic role of CTCF is localizing CENP-E to the centromere (Xiao et al., 2015). CENP-E recruitment can be determined via immunofluorescence and function can be measured by chromosome congression to the metaphase plate (Fig. 3). CENP-E immunofluorescence in the metaphase plate showed no significant change from untreated to 3-day CTCF degradation via 5-Ph-IAA (Fig. 3A,B). By contrast, the inhibition of CENP-E via GSK-923295 for 1 day resulted in a drastic and significant 60% loss of CENP-E intensity in the metaphase plate chromosomes (Fig. 3B). Failure of chromosome congression driven by CENP-E results in polar chromosomes, defined as chromosomes that fail to congress to the metaphase plate and remain near the spindle pole (Barisic et al., 2014). CTCF KD for 3 days resulted in a modest but insignificant increase in the presence of polar chromosomes in metaphase from 4% untreated to 13% in CTCF KD with the majority showing only a single chromatid pair at a spindle pole (Fig. 3C). Treatment with the CENP-E inhibitor GSK-923295 for 1 day resulted in polar chromosomes in 62% of cells in metaphase, the majority of which had more than one sister chromatid pair at the spindle poles (Fig. 3A,C), in agreement with a

previous publication (Bennett et al., 2015). Taken together, the data does not support a role for CTCF in CENP-E recruitment or function.

The disagreement between our data showing no dependence and previous reports of a CENP-E dependence on CTCF might be due to the difference in CTCF loss time. Our data via 3-day CTCF KD is shorter than other publications showing the effects of longer-term loss of CTCF (Xiao et al., 2015; Chiu et al., 2023; Watanabe et al., 2023). Thus, it is possible that the loss of CTCF over time disrupts other centromere and kinetochore structures and/or functions. However, we find that the primary role of CTCF is not CENP-E recruitment but likely a different function in the centromere.

### CTCF is essential to centromere structure and metaphase plate alignment

To determine the role of CTCF in the centromere, we aimed to compare loss of CTCF to loss of cohesin, a known component of the centromere. Partial loss of cohesin has been shown to disrupt the centromere chromatin spring structure and function, while maintaining sister cohesion (Maresca and Salmon, 2009; Carvalhal et al., 2018). RAD21 is an essential subunit of the cohesin ring complex (Rudra and Skibbens, 2013). We used the previously generated CRISPR cell line RAD21-mAID-Clover HCT116 cells (Yesbolatova et al., 2020), and treated them with 357 μM of 5-Ph-IAA for 6 h to partially degrade cohesin, resulting in a 40% knockdown (Fig. 4AB; Fig. S2). We then measured intercentromere distance and metaphase plate width upon CTCF knockdown and RAD21 partial knockdown to determine the role of CTCF in the centromere.

Centromeric structure can be directly measured by determining the intercentromere distance imaged via spinning disc confocal microscopy. Using anti-centromere antibody (ACA), we used immunofluorescence to label and measure the distance between sister centromeres (Fig. 4C). In untreated cells, the intercentromere distance measured between bioriented sister chromatid centromeres in metaphase was 1.08±0.01 μm (mean±s.e.m.). CTCF knockdown for 3 days resulted in a significant increase in intercentromere

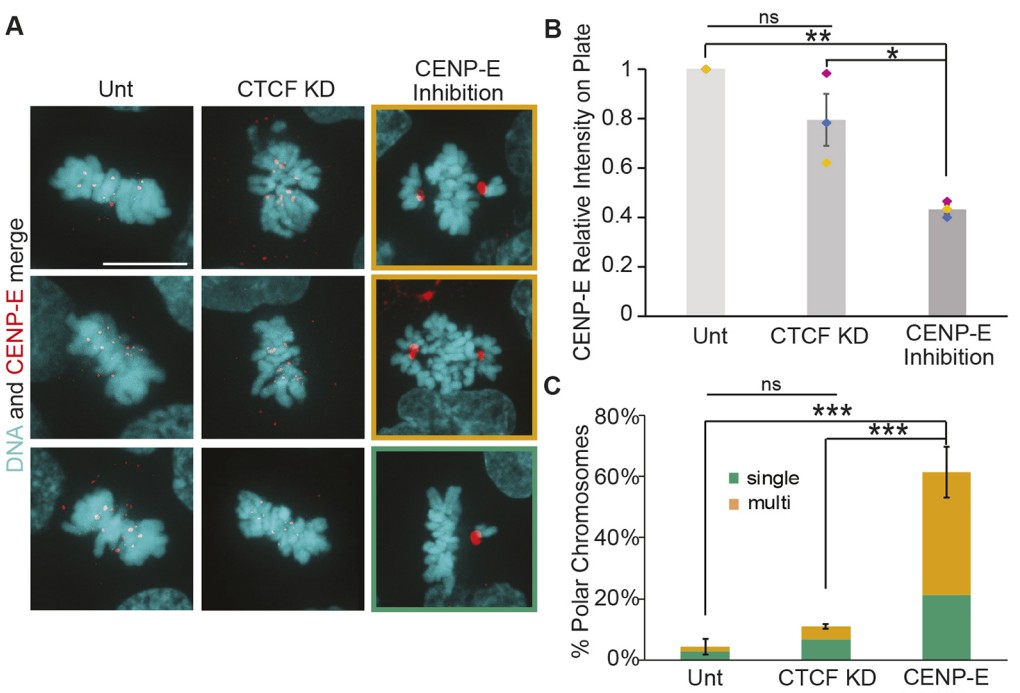

**Fig. 3. CENP-E intensity and chromosome congression remain intact upon CTCF knockdown.** (A–C) Metaphase example images of CENP-E immunofluorescence (red) and DNA (cyan, Hoechst 33342) (A) and graphs of CENP-E intensity in metaphase plate (B) and percentage of spindles with single (green) or multiple (yellow) polar chromosomes (C) in untreated (Unt), CTCF KD via 3-day 5-Ph-IAA treatment, and CENP-E inhibitor GSK-923295. Results in B and C are mean±s.e.m. (three biological replicates with *n*>12 each). *P<0.05; **P<0.01; ***P<0.001; ns, not significant (one-way ANOVAs followed by post hoc Tukey tests). Scale bar: 10 μm..

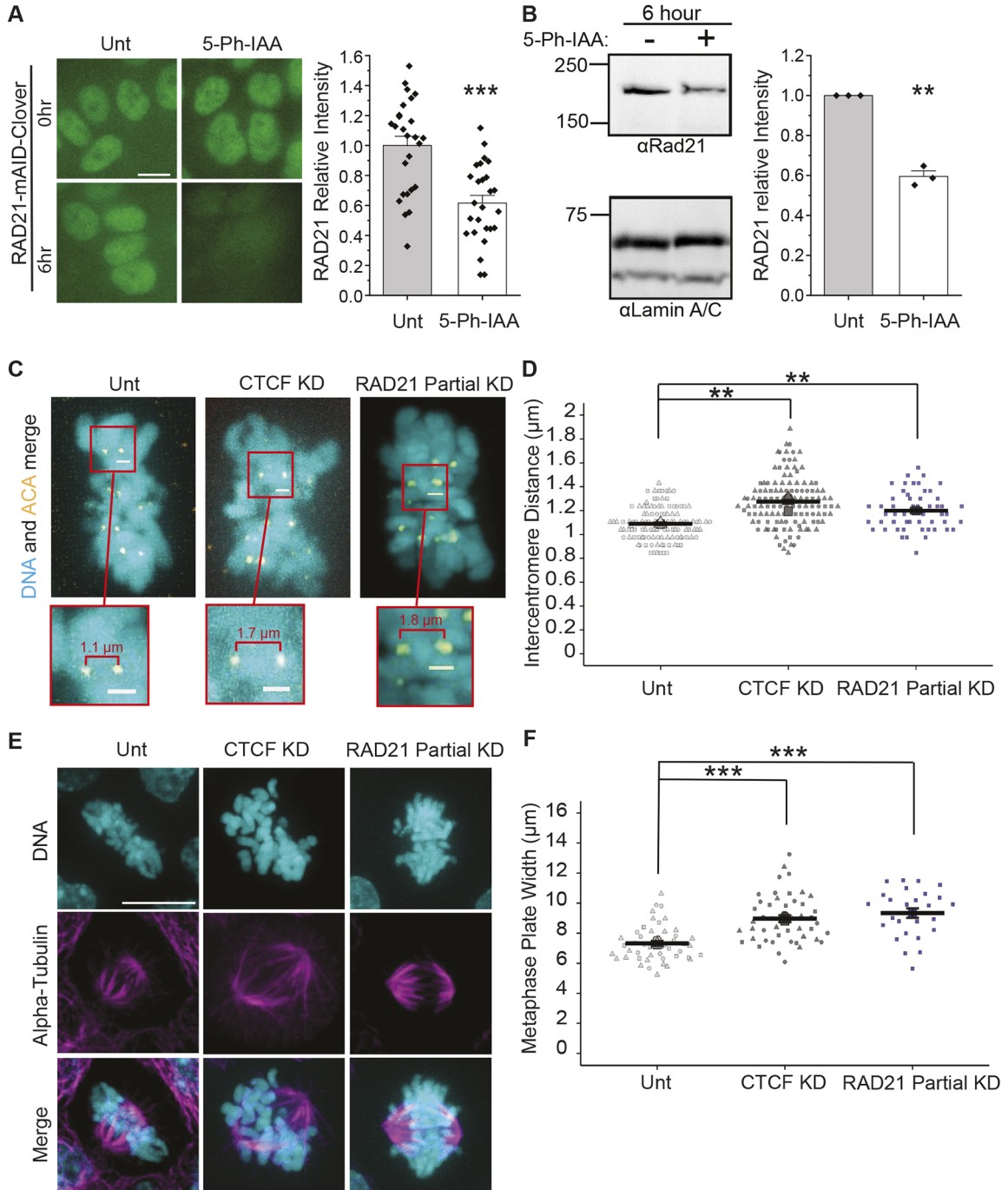

**Fig. 4. Intercentromere distance and metaphase plate width increase with CTCF knockdown and RAD21 partial knockdown.** (A,B) Example fluorescence images (A, *n*=26) and western blots with quantification (B) of RAD21-mAID-Clover relative intensity upon partial knockdown via 357 µM 5-Ph-IAA for 6 h for biological triplicates. Scale bar: 10 µm. Western blots shown are representative of three biological replicates with raw data in Fig. S2. (C,D) Example images of sister centromere pairs DNA (cyan, Hoechst 33342) and centromere immunofluorescence (yellow, ACA) (C) and graph of intercentromere distances in metaphase plates (D) in untreated (Unt), 3-day 5-Ph-IAA (CTCF KD), and RAD21 partial knockdown. Scale bars: 1 µm. (E) Example images of metaphase plates labeled with DNA (cyan, Hoechst 33342) and alpha tubulin immunofluorescence (magenta) in untreated, CTCF KD, RAD21 partial KD. Scale bar: 10 µm. (F) Graph of metaphase plate width. For C–F, three biological replicates *n*>15 each and RAD21 partial KD one replicate *n*=16. Results show mean±s.e.m. **$P<0.01$; ***$P<0.001$ (unpaired two-tailed Student's *t*-tests between untreated and each KD).

distance to 1.26±0.04 µm (Fig. 4D). Similarly, partial cohesin degradation (RAD21 KD) resulted in a significant increase relative to untreated in the intercentromere distance to 1.19±0.02 µm. This data supports the hypothesis that CTCF has a role, along with cohesin, in maintaining the centromere chromatin structure as a spring to resist bioriented microtubule pulling forces.

The centromere spring aids in both tension sensing and alignment of chromosomes in metaphase. We measured the metaphase plate width using spinning disk confocal imaging of DNA labeled via Hoechst 33342. In untreated cells, the metaphase plate width visually appears tight and measured 7.33±0.07 µm (mean±s.e.m.). CTCF knockdown for 3 days resulted in visually disrupted

metaphase plates, which increased significantly in width to 8.96 ±0.08 µm relative to untreated (Fig. 4E,F). Similarly, partial cohesin degradation resulted in a significant increase in metaphase plate width to 9.33±0.31 µm (Fig. 4E,F). Overall, loss of CTCF disrupts chromosome organization on the metaphase plate and increases intercentromere distance, two roles dedicated to the centromere chromatin spring which is also maintained by cohesin.

CTCF is known to be enriched in the centromere (Rubio et al., 2008; Xiao et al., 2015; Del Rosario et al., 2019; Miyata et al., 2021). However, the role of CTCF in the centromere remained unknown until now. Our data provides strong support for CTCF having a role in maintaining centromere structure and function through the intercentromere distance and metaphase plate alignment. Disruption of the kinetochore has been shown to decrease the interkinetochore or centromere distance (Maresca and Salmon, 2009). This outcome is likely due to the inability to attach and biorient sister chromatids to pull them in opposite directions, resulting in less stretching or distance. Considering this, our data show that CTCF is not essential for centromere attachment as the intercentromere distance increased, not decreased. Instead, attached and bioriented centromeres appear to be unable to resist this tension due to a weakened centromere chromatin spring, resulting in elongation. It has been established that cohesin and condensin and other proteins maintain the mammalian centeromere and chromosome structure through both loops (Sen Gupta et al., 2023; Samejima et al., 2025) and the yeast centromere spring, which is composed of loops on loops (Yeh et al., 2008; Stephens et al., 2011, 2013b,a; Lawrimore et al., 2018; Kolbin et al., 2025). Chromatin conformation interactions or chromatin crosslinking has been shown to maintain the chromatin-based nuclear spring constant in nuclei (Stephens et al., 2017; Belaghzal et al., 2021; Strom et al., 2021), showing this structure is well known to provide a resistive forces. Partial loss of cohesin, where sister cohesion is maintained, results in increased inter-kinetochore or centromere distance and mitotic errors (Maresca and Salmon, 2009; Carvalhal et al., 2018; Zielinska et al., 2019). Thus, loss of CTCF recapitulates what is seen upon partial cohesin loss, which we have recapitulated in our data. We provide novel data that the role of CTCF in mitosis is similar to its role in interphase – to aid cohesin in maintaining chromatin structure. These data overall support that CTCF maintains the centromere spring and metaphase plate alignment to aid proper mitotic fidelity and normal nuclear shape post mitosis.

## MATERIALS AND METHODS
### Cell culture
HCT116 WT and HCT116 CTCF-mAID-Clover and RAD21-mAID-Clover cells (Masato T. Kanemaki, National Institute of Genetics, USA) were grown in McCoys 5A medium (Thermo Fisher Scientific, 16600-082) supplemented with 10% fetal bovine serum (FBS; HyClone) and 1% penicillin-streptomycin (Corning) (complete medium). The cells were allowed to grow to 70% confluency on 60 mm dishes before being passaged every 2–3 days.

### Drug treatment
The drug of interest for this paper is 5-Ph-IAA (Sigma-Aldrich, SML3574), the inducer of the AID system. For the CTCF-mAID-Clover cells, cells were plated and allowed to grow on the dish for 24 h before being treated with 10 µM of 5-Ph-IAA for 1–3 days depending on the condition. Medium was aspirated, replaced and re-treated every 24 h to ensure proper functioning of the 5-Ph-IAA. For the RAD21-mAID-Clover cells, cells were plated and allowed to grow on the dish for 24 h before being treated with 357 µM of 5-Ph-IAA for 6 h.

GSK-923295 (APExBIO, a3450) was used to represent CENP-E inhibition phenotypes. Cells were plated, allowed to grow for 24 h, and then treated with 10 nM of GSK for 1 day before imaging.

### Time lapse imaging and analysis
Images were acquired with Nikon Elements software on a Nikon Instruments Ti2-E microscope, Orca Fusion Gen III camera, Lumencor Aura III light engine, TMC CLeanBench air table, with 40× air objective (NA 0.75, W.D. 0.66, MRH00401). Live-cell time lapse imaging was undertaken with the Nikon Perfect Focus System and Okolab system at 37°C, with humidity and at 5% $CO_2$ (stage top incubator, H301).

For rapid degradation imaging, cells were plated in four-well imaging dishes (Cellvis, D35C4-20-1.5-N) and allowed to grow for 1 day. The cells were then treated with 1:1000 SPY650-DNA (Thermo Fisher Scientific, NC2096299) and 5-Ph-IAA, and imaging began after 45 min. Images were taken in 7-min intervals for 3 h with 5 fields of view per condition. Cells were imaged using the Cy5 channel to visualize the DNA, and FITC to visualize the CTCF-mAID-Clover. ROIs were drawn around nuclei according to the DNA staining, and the FITC intensities were taken throughout the time lapse. Data was transferred to excel and background subtracted. The percentage intensity at the various time points was compared to the intensity at $t$=0.

For mitotic failure imaging, cells were plated in four-well imaging dishes and allowed to grow for 1 day before being treated with 5-Ph-IAA for the appropriate number of days. Cells were then treated with SPY650-DNA and imaging commenced after 45 min. Images were taken in 10-min intervals for 16 h with seven fields of view per condition and a 3-step z-stack of 2.5 µm steps to cover 5 µm. Cells were imaged using the Cy5 channel to visualize DNA as well as the consolidation of DNA associated with the steps of mitosis.

Analysis of mitotic duration was calculated by measuring the time from the condensation of the DNA to the abscission of the two nuclei, or in the case of failed mitosis the eventual expansion of the DNA. The number of frames from beginning to end was multiplied by the length of the frame (10 min), and this data was noted in Excel (Microsoft).

Mitotic failure was categorized in one of the five categories (shown in Fig. 1E) and noted in Excel, and the activity of the nuclei were confirmed to match to each category by observing their motion for 1 h post mitosis. Categorization was performed by a researcher who was aware of the experimental conditions.

Area and circularity for each condition was calculated at the beginning ($t$=0) of each time lapse, and when the treatment was exactly 48 or 72 h (corresponding to 2 and 3 days). To do this, a Bezier ROI was drawn around each nucleus that did not overlap any other nucleus, converted into binary, and then the circularity measurement was extracted and pasted into Excel where the data from all three replicates were compiled. For post mitotic circularity, the nuclei were allowed to exit mitosis, and then at 10 frames (100 min) post mitosis an ROI was hand drawn over any resulting individual daughter nuclei. The ROI was converted to binary, and the data was pasted into Excel.

### Immunofluorescence
Cells were plated in eight-well imaging dishes (Cellvis, C8-1.5H-N) and allowed to grow for 1 day before treatment with 5-Ph-IAA for multiple days. Cells were then fixed in 4% paraformaldehyde (Electron Microscopy Sciences), washed with phosphate-buffered saline (PBS; Corning) three times for 5 min each. Cells were permeabilized with 0.1% Triton X-100 (Promega) for 15 min, then with 0.06% Tween 20 (PBS-T) for 5 min, and then three washing steps with PBS were completed. Cells were then blocked in 2% bovine serum albumin (BSA; Thermo Fisher Scientific) in PBS for 1 h at room temperature.

Primary antibody dilutions were created using BSA in PBS. The primary antibodies used were: anti-CTCF antibody (rabbit, 2899, Cell Signaling Technologies) at 1:100; anti-CENP-E antibody (mouse, 39619, Active Motif) at 1:1000; and anti-centromere antibody (ACA) human (HCT-0100 Immunovision) at 1:4000. Primary antibodies were added, and cells were incubated overnight at 4°C. Cells were then washed three times with PBS. Secondary antibody dilutions were created with BSA in PBS and cells were incubated with concentrations of 1:1000 for 1 h on a shaker at room temperature. The secondary antibodies used were: goat anti-human-IgG conjugated to Alexa Fluor 555 (A21433 Invitrogen), goat anti-mouse-IgG conjugated to Alexa Fluor 647 (4410, Cell Signaling Technologies), and goat anti-rabbit-IgG conjugated to Alexa Fluor 647 (4414, Cell Signaling Technologies). Cells were then washed three times with PBS.

Journal of Cell Science

The cells were then stained with 1:10,000 concentration of Hoechst 33342 (H3570, Thermo Fisher Scientific) for 10 min before three more PBS washing steps. The cells were then mounted using 30 µl Gold Antifade per well (P36930, Thermo Fisher Scientific).

### Immunofluorescence imaging and analysis

Spinning disc confocal images were acquired with Nikon Elements software on a Nikon Instruments Ti2-E microscope with Crest V3 Spinning Disk Confocal, Orca Fusion Gen III camera, Lumencor Celesta light engine, a TMC CleanBench air table, with a Plan Apochromat Lambda 100× oil immersion objective lens (NA 1.45, W.D. 0.13 mm, F.O.V. 25 mm, MRD71970) as previously described (Pho et al., 2023).

The metaphase plate stained by Hoechst 33342 was visualized in DAPI and imaged with a 100× oil objective. The width of the metaphase plate was calculated by creating a maximum projection image in the $Z$-plane with DNA, and then measuring the distance between the edges of DNA using the length tool in annotations and measurements in NIS Elements software. This measurement includes the metaphase tail that is present in most plates and also includes any chromosomes that were not successfully connected to the metaphase plate during the transition from prophase to metaphase. This ROI was converted into a binary image, where its metaphase plate width measurement was taken and compiled into Excel. Categorization was performed by a researcher who was aware of the experimental conditions.

The centromeres were visualized through TRITC after staining of anti-centromere antibody (ACA). They were imaged with a $Z$-stack of 0.3 µm steps for 5 µm, and imaged with a 100× oil objective with a spinning disk confocal microscope with settings of 60% fox excitation light power, 300 ms of exposure time and 12-bit camera settings. The intercentromere distance was measured by cropping an image of a metaphase plate, drawing an ROI where only background fluorescence was present in both DAPI and TRITC channels. This ROI was set as a background ROI and then used to background subtract the image. A line scan was then drawn through two foci that were in the same plane on the metaphase plate and which had a visible chromosome between them. The distance between the peak intensities of the two foci was calculated and pasted in Excel as the intercentromeric distance for that one pair. This was completed 1–3 times per metaphase plate.

CENP-E was visualized through CY5 and imaged with a 100× oil objective with a spinning disk confocal microscope with settings of 99% fox excitation light power, 350 ms of exposure time and 12-bit-sensitive camera settings. The metaphase plate was cropped in NIS Elements software, and the background signal was subtracted by drawing an ROI, setting it as a background ROI, and subtracting that signal from the image. A maximum projection in the $Z$-plane was created with the DNA and CENP-E signals, and then a bounding box was drawn around the main metaphase plate (excluding polar chromosomes and tails). The intensity of CENP-E in each plate was determined and analyzed in Excel.

### Western blotting

HCT116 cells expressing CTCF-mAID-Clover or RAD21-mAID-Clover or were plated into a six-well plate 16–24 h ahead of drug treatment. Cells were then treated with 10 µM 5-Ph-IAA for 4 h, 24 h or 72 h (CTCF-mAID-Clover), or 357 µM 5-Ph-IAA for 6 h (RAD21-mAID-clover). All treated wells had an accompanying untreated well. Cells were washed with Hanks' balanced salt solution (Gibco, 14025-076), released with trypsin (Gibco), collected into complete medium, and pelleted. Pellets were washed with PBS, before storing at −80°C for later lysis.

For CTCF-mAID-Clover, cell pellets were resuspended in lysis buffer (50 mM Tris-HCl pH 8.0, 150 mM NaCl, 5 mM EDTA, 1% IGEPAL CA-630, 0.1% SDS, 1 mM PMSF and a Roche Protease Inhibitor Cocktail tablet EDTA free). Lysate was incubated on an end-over-end rotator at 4°C for 15 min, before centrifuging at 13,000 RPM (20,821 $g$) for 5 min at 4°C. Supernatant was collected, protein concentration was assessed with Protein Assay Dye Reagent (Bio-Rad, 5000006), 5× SDS sample buffer was added, and the sample was boiled for 5 min.

For RAD21-mAID-Clover, nuclear fractions were collected (Suzuki et al., 2010; Senichkin et al., 2021). Pellets were resuspended in PBS plus 0.1% IGEPAL CA-630, vortexed, then pelleted at 5000 RPM (3080 $g$) for 10 min at 4°C. Pelleted nuclei were then resuspended in lysis buffer,

vortexed, and incubated on ice for 10 min with intermittent vortexing, before centrifugation at 13,000 RPM (20,821 $g$) for 5 min at 4°C. Supernatant was collected, protein concentration was taken with Protein Assay Dye Reagent, and 5× SDS sample buffer was added.

Equivalent protein samples were loaded onto 10% acrylamide gels made in-house for SDS-PAGE. Gels were run out completely, before transferring onto nitrocellulose membrane with Bio-Rad Trans-blot Turbo with the 'High MW' program for 10 min. Antibody were diluted in 5% non-fat dry milk dissolved in Tris-buffered saline (TBS) with 0.1% Tween-20. Primary antibodies used were: rabbit anti-CTCF antibody (Cell Signaling, 2899) at 1:2500, rabbit anti-RAD21 antibody (Cell Signaling, 4321) at 1:2500, mouse anti-α-tubulin antibody (Sigma-Aldrich, T9026) at 1:5000, and mouse anti-Lamin A/C antibody (Cell Signaling, 4777) at 1:2500. Secondary antibodies used were: peroxidase-conjugated donkey anti-rabbit-IgG (Jackson ImmunoResearch, 711-035-152) at 1:5000 and peroxidase-conjugated donkey anti-mouse-IgG at 1:2500 (Jackson ImmunoResearch, 715-035-150). Membrane was first probed with anti-CTCF or anti-RAD21 antibody overnight at 4°C. Membrane was incubated with secondary antibody at room temperature for 45 min. Membrane was washed three times for 5 min each with TBS plus 0.1% Tween 20 after primary and secondary antibody incubations. Membranes were developed with Immobilon Western Chemiluminescent HRP substrate (Millipore Sigma, WBKLS0500) and imaged with Syngene G-box. For loading controls, membranes were washed, re-blocked, then re-probed with mouse anti-α-tubulin or mouse antibody anti-lamin A/C antibody for 1 h at room temperature, followed by secondary antibody for 45 min, washing with TBS plus 0.1% Tween-20 after each antibody incubation.

Band intensities were normalized against tubulin or lamin A for CTCF or RAD21, respectively. Quantifications were performed in ImageJ. Data was processed in Excel and figures were compiled in Adobe Illustrator.

### Statistics

After data was compiled, Shapiro–Wilks normality tests as well as Levene tests for homogeneity of variance were conducted with RStudio. If the datasets did not pass this test, a Kruskal–Wallis test was run followed by a Dunn's test. If the data sets passed these normality tests with a $P>0.05$, the data sets were tested for statistical significance using a one-way ANOVA test. If the resulting $P$ value was $>0.05$, then the data was not statistically significant. If the resulting $P<0.05$, then the data was statistically significant, and a post hoc Tukey test was run to test for significance between each group. A group was considered significantly different from another if the $P<0.05$. If the data did not have three replicates, a two-tailed unpaired Student's $t$-test was run instead. The statistical tests run on each data set are noted in each figure legend. To summarize, one-way ANOVA and post hoc Tukey tests were run on Figs 1D, 2A,B,E, 3B,C, 4D and Fig. S1E. Kruskal–Wallis and Dunns tests were run on Figs 2D and 4B. two-tailed unpaired Student's $t$-tests were run on Fig. S1B–D and S1F.

### Acknowledgements
We would like to thank Masato T. Kanemaki for kindly sharing the CTCF-mAID-Clover and RAD21-mAID-Clover cell lines. We would like to thank Kelsey Prince for feedback and support as a great lab mate.

### Competing interests
The authors declare no competing or financial interests.

### Author contributions
Conceptualization: T.L.; Data curation: E.W.; Formal analysis: E.W., T.L.; Funding acquisition: T.J.M., A.D.S.; Investigation: E.W., T.L.; Methodology: E.W.; Project administration: A.D.S.; Validation: E.W., T.L.; Visualization: E.W., T.L.; Writing – original draft: A.D.S.; Writing – review & editing: E.W., T.J.M.

### Funding
This work was supported by NIH (National Institutes of Health) NIGMS grant Maximizing Investigators' Research Awards R35GM154928 to A.D.S. and R35GM156188 to T.J.M. T.L. was supported by T32GM135096 as a fellow in the UMass Biotechnology Training Program. Open Access funding provided by University of Massachusetts Amherst. Deposited in PMC for immediate release.

**Data and resource availability**
All figure data is available as in a public repository in FigShare (doi:10.6084/m9.figshare.29197922.v1). All other relevant data and details of resources can be found within the article and its supplementary information.

**Peer review history**
The peer review history is available online at https://journals.biologists.com/jcs/lookup/doi/10.1242/jcs.264181.reviewer-comments.pdf

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
