## [Peer Review File · Journal of Cell Science]

CTCF maintains centromere function and mitotic fidelity

Erin Walsh, Thomas Laskarzewski, Thomas J. Maresca and Andrew D. Stephens

DOI: 10.1242/jcs.264181

Editor: Renata Basto

Review timeline

Original submission:	5 June 2025
Editorial decision:	28 July 2025
First revision received:	24 October 2025
Editorial decision:	25 November 2025
Second revision received:	17 December 2025
Accepted:	22 December 2025

Original submissionFirst decision letter

MS ID#: jcs.264181

MS TITLE: CTCF maintains pericentromere function and mitotic fidelity

AUTHORS: Erin Walsh; Andrew D Stephens

ARTICLE TYPE: Short Report

Dear Dr Stephens,

We have now reached a decision on the above manuscript.

To see the reviewers' reports and a copy of this decision letter, please go to:

As you will see, the reviewers raise a number of substantial criticisms that prevent me from accepting the paper at this stage. They suggest, however, that a revised version might prove acceptable, if you can address their concerns. If you think that you can deal satisfactorily with the criticisms on revision, I would be pleased to see a revised manuscript. We would then return it to the reviewers.

Reviewer 1*Advance summary and potential significance to field*

The authors establish CTCF-mAID-Clover cell line and find that mitosis is significantly affected only after 3 days of CTCF depletion, consistent with previous findings that the 3D genome organization is affected only after the same time of CTCF depletion. They reveal that the majority of mitotic defects are constituted by anaphase bridges. They also reveal, that CTCF depletion for 3 days causes loss of post-mitotic, but not population nuclear circularity. Furthermore, the authors show that depletion of CTCF does not mimic the effect of CENP-E inhibition, the latter causing polar chromosomes. In addition to this, CTCF depletion increases the interkinetochore distance and metaphase plate width, which phenocopies the partial depletion of cohesin subunit RAD21.

The study suggests the cooperation of CTCF and cohesin in maintaining mitotic sister centromere tension and alignment of metaphase plate in human cells and is thus of broad interest to the cell biology community.

The data presented is very solid and the conclusions are valid, however before the paper is accepted there are a few points to be considered:

- a) The extent of CTCF and RAD21 depletion needs to be presented by semi-quantitative western blot.
- b) Cohesin depletion was shown to affect the 3D structure of mitotic chromosomes (PMID: 40208986), it would be good to discuss it accordingly.

Minor suggestions for text corrections:

Page 4 Lines 36-41 „The most prominent anaphase phenotype across conditions was an anaphase bridge between the daughter nuclei (orange, Fig. 1E). Less frequently, anaphase failed to split the DNA, resulting in all chromosomes being in one daughter nucleus (blue, Fig. 1E), or binucleation occurred in which daughter nuclei separated but remained in one cell (magenta, Fig. 1E).”

Page 5 lines 10-16 "Anaphase bridges can lead to systemic genomic instability (Rodriguez-Muñoz et al., 2022). Concatenations that occur during DNA replication are usually decatenated by topoisomerase II aided by cohesin, condensin, and mitotic spindle microtubule tension that resolve sister chromatids (Finardi et al., 2020)."

Page 6 lines 48-52 "Treatment with CENP-E inhibitor GSK 923295 for 1 day resulted in polar chromosomes in 62% of cells in metaphase, the majority of which had more than one sister chromatid pair at the spindle poles (Fig. 3, A and C),"

Page 7 lines 50-55 "To recapitulate those findings and compare to CTCF KD, we used previously generated CRISPR cell line RAD21-mAIDClover HCT116 cells (Yesbolatova et al., 2020) and treated it with 357 μ M of 5-Ph-IAA for 6 hours to partially degrade cohesin (Supplemental Figure 1F)."

Reviewer 2

Advance summary and potential significance to field

Comments to ms draft entitled "CTCF maintains pericentromere function and mitotic fidelity" by Walsh et al. submitted to Journal of Cell Science.

In this manuscript, the authors examine the mitotic role of CTCF, which has been reported to localize to pericentromeres. The authors conclude that CTCF facilitates chromatin looping rather than recruiting CENP-E to the kinetochore as described previously (Xiao et al., 2015), based on: a) CTCF knockdown induces anaphase separation failures and abnormal nuclear shape; b) CTCF knockdown does not induce polar chromosomes, unlike CENP-E inhibition, challenging the idea that CTCF recruits CENP-E to the centromere; c) inter-centromere distance and metaphase plate width were increased in CTCF knockdown-cells, suggesting that CTCF is required to maintain the pericentromere structure to withstand microtubule-pulling force during mitosis.

These results point to the role of CTCF in mitosis which has been controversial. The experiments are straightforward and the results are interesting. However, I see several issues that authors would need to further address to improve the manuscript as detailed below.

Comments for the author

Major comments:

#1 The advantage of AID system is that it allows rapid degradation of the target protein. However, the experiment in figure 1, cellular phenotypes are examined after 3 days of 5-Ph-IAA treatment. In line with authors' description in the discussion of figure 3, a long-term knockdown of up to 3 days

may allow for secondary effects and prevent direct interpretation. According to figure 1B, CTCF seems to be largely degraded already after 3 hr. Does the 3 hr degradation of CTCF cause any abnormalities in mitosis? If yes, those results demonstrate better that CTCF plays a primary role in preventing mitotic abnormalities. If that is not the case, a side-by-side quantification of CTCF intensities after 3 hr and 3 days should help interpret why the phenotypes emerges after 3 days. Please address these possibilities.

#2 The authors show that nuclear circularity was significantly decreased after mitosis following 3 days of CTCF knockdown. However, the authors also show that total population of cells with decreased nuclear circularity remained unchanged (Figure 2 D, E). Provided that a non-negligible fraction of cells underwent mitosis over the 3-day period, it seems inconsistent that total cell population remains unchanged after CTCF knockdown. Please clarify.

#3 One of the novel findings of this paper is that CTCF maintains pericentromere structure. However, I am not sure if this conclusion is directly supported by the increased inter-centromere distance (Figure 4A, B). My understanding is that pericentromere is a distinct heterochromatic domain formed adjacent to centromeres, where kinetochores are assembled and microtubule-pulling force is applied in mitosis. How might structure of the pericentromere influence the rigidity of centromere? I don't think this link is well understood. I consider it important to distinguish pericentromeres from centromeres, and address the potential connection between them.

Minor comments :

#4 The abstract would be easier to read, if pericentromeric localization of CTCF is described earlier, rather than being mentioned in its last part.

#5 The figure 1E could be self-explanatory if the authors describe which colors indicate which kinds of mitotic errors.

#6 Readers in general are not familiar with terms such as 'ESC' or '5C' (in discussion of Figure 1).

#7 In Figure 4, the authors conducted a RAD21 knockdown experiment. The authors may want to provide some background and explain the specific aim of this experiment, as it seems rather abrupt at present.

First revision

Author response to reviewers' comments

Reviewer 1: The authors establish CTCF-mAID-Clover cell line and find that mitosis is significantly affected only after 3 days of CTCF depletion, consistent with previous findings that the 3D genome organization is affected only after the same time of CTCF depletion. They reveal that the majority of mitotic defects are constituted by anaphase bridges. They also reveal, that CTCF depletion for 3 days causes loss of post-mitotic, but not population nuclear circularity. Furthermore, the authors show that depletion of CTCF does not mimic the effect of CENP-E inhibition, the latter causing polar chromosomes. In addition to this, CTCF depletion increases the interkinetochore distance and metaphase plate width, which phenocopies the partial depletion of cohesin subunit RAD21.

The study suggests the cooperation of CTCF and cohesin in maintaining mitotic sister centromere tension and alignment of metaphase plate in human cells and is thus of broad interest to the cell biology community.

The data presented is very solid and the conclusions are valid, however before the paper is accepted there are a few points to be considered:

a) The extent of CTCF and RAD21 depletion needs to be presented by semi-quantitative western blot.

In the revised manuscript we provide new data of semi-quantitative western blots for degradation of CTCF and RAD21. We include new Western blot data in Figure 1C of CTCF degradation over 4 hours, 1 day and 3 days shows a $\geq 95\%$ loss of signal. We also include new data via a Western blot of CTCF KD over 4 hours in biological replicates which shows $> 85\%$ knockdown in Supplemental Figure 1A. We also included new data of Western blots of RAD21 partial depletion resulted in a 40% loss for biological triplicates which agrees with original immunofluorescence data - both are now added to Figure 4 as panels A and B. We have included the raw western blots in Supplemental Figure 2.

b) Cohesin depletion was shown to affect the 3D structure of mitotic chromosomes (PMID: 40208986), it would be good to discuss it accordingly.

We thank the reviewer for reminding us of this important work which we now cite in the revised manuscript introduction and final discussion section.

Minor suggestions for text corrections:

Page 4 Lines 36-41 „The most prominent anaphase phenotype across conditions was an anaphase bridge between the daughter nuclei (orange, Fig. 1E). Less frequently, anaphase failed to split the DNA, resulting in all chromosomes being in one daughter nucleus (blue, Fig. 1E), or binucleation occurred in which daughter nuclei separated but remained in one cell (magenta, Fig. 1E)."

We thank the reviewer for their feedback. We have made the suggested revision.

Page 5 lines 10-16 "Anaphase bridges can lead to systemic genomic instability (Rodriguez-Muñoz et al., 2022). Concatenations that occur during DNA replication are usually decatenated by topoisomerase II aided by cohesin, condensin, and mitotic spindle microtubule tension that resolve sister chromatids (Finardi et al., 2020)."

We thank the reviewer for their feedback. We have made the suggested revision.

Page 6 lines 48-52 "Treatment with CENP-E inhibitor GSK 923295 for 1 day resulted in polar chromosomes in 62% of cells in metaphase, the majority of which had more than one sister chromatid pair at the spindle poles (Fig. 3, A and C),"

We thank the reviewer for their feedback. We have made the suggested revision.

Page 7 lines 50-55 "To recapitulate those findings and compare to CTCF KD, we used previously generated CRISPR cell line RAD21-mAIDClover HCT116 cells (Yesbolatova et al., 2020) and treated it with 357 μM of 5-Ph-IAA for 6 hours to partially degrade cohesin (Supplemental Figure 1F)."

We thank the reviewer for their feedback. We have revised this and the surrounding text to clarify the goal of using partial cohesin knockdown and how it was accomplished.

Reviewer 2: SUMMARY OF THE ADVANCE MADE IN THIS PAPER AND ITS POTENTIAL SIGNIFICANCE TO THE FIELD

Comments to ms draft entitled "CTCF maintains pericentromere function and mitotic fidelity" by Walsh et al. submitted to Journal of Cell Science.

In this manuscript, the authors examine the mitotic role of CTCF, which has been reported to localize to pericentromeres. The authors conclude that CTCF facilitates chromatin looping rather than recruiting CENP-E to the kinetochore as described previously (Xiao et al., 2015), based on: a) CTCF knockdown induces anaphase separation failures and abnormal nuclear shape; b) CTCF knockdown does not induce polar

chromosomes, unlike CENP-E inhibition, challenging the idea that CTCF recruits CENP-E to the centromere; c) inter-centromere distance and metaphase plate width were increased in CTCF knockdown-cells, suggesting that CTCF is required to maintain the pericentromere structure to withstand microtubule-pulling force during mitosis.

These results point to the role of CTCF in mitosis which has been controversial. The experiments are straightforward and the results are interesting. However, I see several issues that authors would need to further address to improve the manuscript as detailed below.

SUGGESTIONS TO AUTHORS

Major comments:

#1 The advantage of AID system is that it allows rapid degradation of the target protein. However, the experiment in figure 1, cellular phenotypes are examined after 3 days of 5-Ph-IAA treatment. In line with authors' description in the discussion of figure 3, a long-term knockdown of up to 3 days may allow for secondary effects and prevent direct interpretation. According to figure 1B, CTCF seems to be largely degraded already after 3 hr. Does the 3 hr degradation of CTCF cause any abnormalities in mitosis? If yes, those results demonstrate better that CTCF plays a primary role in preventing mitotic abnormalities. If that is not the case, a side-by-side quantification of CTCF intensities after 3 hr and 3 days should help interpret why the phenotypes emerges after 3 days. Please address these possibilities.

The reviewer is asking for two things: 1) clarification on mitotic failure rates after 3-hour degradation of CTCF and 2) comparison of short and multi-day long degradation of CTCF.

- 1) Mitotic failure rates were addressed in the original manuscript in Figure 1D in which imaging of mitosis was done for 4-20 hours of auxin treatment, which is denoted as 1-day CTCF KD. This allows us to determine mitotic failure rates upon rapid degradation (> 80%) after 3 hours in auxin. We find that $3\pm 1\%$ mitotic failure in wild type vs. 1 days of CTCF KD with $7\pm 2\%$ mitotic failure, which is not statistically significant. 2 day CTCF KD (28 -44 hours) also shows no statistically significant increase. 3 days of CTCF KD are needed to see an significant increase in mitotic failure rates. Thus, short term degradation of CTCF does not impact mitotic failure significantly but long term does in agreement with our previous publication (Chiu et al., 2023 Chromosoma). We discuss that this is likely due to CTCFs role in maintenance and not establishment.
- 2) CTCF intensity 3 hours to multi days knockdown via 5-Ph-IAA were quantified via fluorescence in the original manuscript and shown to have similar levels of knockdown which were > 80% for 3 hours (Figure 1A and B) and > 60% for 2 day CTCF KD (Supplemental Figure 1A). Similar to Reviewer #1s request, we have provided Western blot new data to the revised manuscript in Figure 1C showing that auxin treatment to degrade CTCF for 4 hours, 1 day and 3 days provides similar drastic knockdown > 95%. CTCF KD in 4 hours using 5-Ph-IAA western blots show similar > 85% knockdown via newly added data in Supplemental Figure 1A.

#2 The authors show that nuclear circularity was significantly decreased after mitosis following 3 days of CTCF knockdown. However, the authors also show that total population of cells with decreased nuclear circularity remained unchanged (Figure 2 D, E). Provided that a non-negligible fraction of cells underwent mitosis over the 3-day period, it seems inconsistent that total cell population remains unchanged after CTCF knockdown. Please clarify.

The reviewer is asking for clarity on why the post-mitotic nuclear circularity decreases but not the whole nuclear circularity population for 3 days CTCF KD. In short, the post mitotic circularity is a small portion of nuclei which is especially measured 1.5 hours after anaphase onset. The whole cell population nuclear circularity would include roughly half that went through mitosis and have that had not. It would stand to reason that most nuclei will have gone through mitosis from day 3 to day 4 of CTCF KD. We provide new data in the revised manuscript showing that by 4 days of CTCF KD the whole population nuclear circularity population does decrease

significantly from 0.935 in control vs. 0.915 in 4 day CTCF (Supplemental Figure 1F).

#3 One of the novel findings of this paper is that CTCF maintains pericentromere structure. However, I am not sure if this conclusion is directly supported by the increased inter-centromere distance (Figure 4A, B). My understanding is that pericentromere is a distinct heterochromatic domain formed adjacent to centromeres, where kinetochores are assembled and microtubule-pulling force is applied in mitosis. How might structure of the pericentromere influence the rigidity of centromere? I don't think this link is well understood. I consider it important to distinguish pericentromeres from centromeres, and address the potential connection between them.

The reviewer is asking for clarification. In short, the centromere structure can be determined via CENPA labeling and pericentromere structure is best measured via interkinetochore distance between the CENPA labeled centromeres. Genomically the centromere is a smaller portion of the chromosome and is surrounded by drastically larger pericentromeric chromatin region on both sides (Yeh et al 2007 and Gupta et al 2023). In yeast the point centromere is easily < 1 kb while the pericentromere is 50 KB while in mammals many centromere connections in a chromosome span 3-5 MB while the pericentromere is ~20MB. The centromere labeling with ACA to recognize CENPA in our data showed no overall change in centromere dot structure as both untreated and knockdowns maintained a diffraction limited spot. However, the distance between the centromere labels, the pericentromere, increased significantly as shown in Figure 4. This agrees with recent publication showing that centromeres labeled by CENPA are reported to be strong and thus do not stretch. However, the pericentromere built of loops on loops labeled by CENPB are less strong and more susceptible to stretching (Biggs et al. 2025 MBoC). We have revised the discussion to reflect these points in the revised manuscript.

A smaller perturbation of the pericentromere would reflect in interkinetochore distance (a weaker/longer spring) but not centromere structure as the connections to microtubules are still grouped and coordinated, an example is CTCF KD. A drastic perturbation such as complete loss of cohesion has been reported to disrupt both centromere structure visualized by size and splitting of a single sister centromeres labeled via CENPA as well as increased distance between sister centromere foci (Zielinska et al. 2019).

In the original manuscript we discuss at length that the pericentromere is composed of loops on loops acting like spring to resist bio-orientation spindle forces that aid alignment and faithful segregation. The literature is quiet robust to support this statement as we have cited multiple publications detailing this structure. Our other work on chromatin mechanics in the cell nucleus detail that chromatin conformation looping and chromatin crosslinks provide mechanical spring-like resistance (Strom et al. 2021 eLife and Belaghzal et al., 2021 Nat Gen). This tension resisting of the pericentromere is necessary to align and coordinate the 15-20 centromere/kinetochores attachments per mammalian chromosome or whole yeast mitotic spindle.

Minor comments :

#4 The abstract would be easier to read, if pericentromeric localization of CTCF is described earlier, rather than being mentioned in its last part.

We have revised the manuscript to made this change: “In mitosis the duplicated genome is aligned and accurately segregated between daughter nuclei. CTCF is a chromatin looping protein that localizes to the pericentromere in mitosis with an unknown role.”

#5 The figure 1E could be self-explanatory if the authors describe which colors indicate which kinds of mitotic errors.

We appreciate the reviewer's feedback and have add a description to the figure llegend: Mitotic failure phenotypes include no anaphase separation of the DNA (blue), anaphase bridge (orange), binucleation (purple), mitotic exit (yellow), and tripolar spindle (gray).

#6 Readers in general are not familiar with terms such as 'ESC' or '5C' (in discussion of Figure 1).

We have revised the manuscript to make this change: “Two days of CTCF degradation in **embryonic stem cells** are required to affect genome organization measurements (Nora *et al.*, 2017).”

#7 In Figure 4, the authors conducted a RAD21 knockdown experiment. The authors may want to provide some background and explain the specific aim of this experiment, as it seems rather abrupt at present.

We have revised the manuscript to make this change:” **To validate that CTCF has a pericentromere function, we aimed to compare loss of CTCF to loss of cohesin a known component of the pericentromere. Partial loss of cohesin has been shown to have a role in determining the pericentromere chromatin spring structure and function (Carvalho *et al.*, 2018), while maintaining sister cohesion. RAD21 is an essential subunit of the cohesin ring complex (Rudra and Skibbens, 2013). We used previously generated CRISPR cell line RAD21-mAID-Clover HCT116 cells (Yesbolatova *et al.*, 2020). To partially degrade cohesin we treated RAD21-mAID-Clover HCT116 cells with 357 μM of 5-Ph-IAA for 6 hours (Supplemental Figure 1F).”**

Second decision letter

MS ID#: jcs.264181R1

MS TITLE: CTCF maintains pericentromere function and mitotic fidelity

AUTHORS: Erin Walsh; Thomas Laskarzewski; Thomas J Maresca; Andrew D Stephens

ARTICLE TYPE: Short Report

Dear Dr Stephens,

As you will see, the revised version of your manuscript has been seen by the original reviewers. Reviewer 1 is satisfied with the revision, but reviewer 2 raise a number of substantial criticisms that prevent me from accepting the paper at this stage. I am therefore asking whether you think that you can address these concerns. If you think that you can deal satisfactorily with the criticisms on revision, I would be pleased to see a revised manuscript.

Comments from the Reviewers:

Reviewer 1: The revisions have significantly improved the paper and new western blots carefully address the levels of the respective proteins over depletion times.

I would be happy to see this paper accepted in Journal of Cell Science.

Reviewer 2: The authors have addressed the major comments of the reviewer as follows:

#1 Time required for the emergence of mitotic defects:

The authors explicitly mentioned that mitotic defects are seen only after 3 days of CTCF knockdown, and provided a reasonable interpretation that CTCF is involved in the maintenance of the pericentromere but not in its establishment.

#2 Loss of nuclear circularity phenotype:

Admittedly, several major issues arose that prevented me from understanding this experiment. First, I am confused with the authors' apparently inconsistent explanations: In the revision the authors describe that, for the 3-day knockdown experiment, "the whole cell population nuclear circularity would include roughly half that went through mitosis and have that had not". However,

in the original manuscript, in the experiment comparing 3-day and longer (more than 3 days) knockdown, the authors explained that cells underwent 2-3 cell cycles in 3 days.

Second, in the revision the authors claim that "most nuclei will have gone through mitosis from day 3 to day 4 of CTCF knockdown". It reads that the authors predicts that the half of cell population underwent first mitosis after the knockdown during this period. But then one needs to assume that the cell cycle went extremely slow after the CTCF knockdown. If this was the case, one would expect to see a considerable change in the nuclear circularity of the whole population in 4-day CTCF knockdown. What the authors saw was a slight decrease, 0.935 in control versus 0.915 in knockdown (revised Supplemental Figure 1F), but this change is way too small if the second half of cells experienced mitosis during this period.

#3 CTCF-mediated pericentromere maintenance:

I disagree with the authors' view that "the distance between the centromere labels the pericentromere". A general understanding is that the centromere is the chromosomal domain that can be determined "between" the CENP-A nucleosomes, not restricted to CENP-A nucleosomes per se, and the pericentromere is the chromosomal domain adjacent to the centromere. Thus, interkinetochore distance is widely used as a readout for the integrity of centromere structure. The authors added one paragraph and discussed this issue along with the Zielinska et al. (2019) paper. However this reference does not describe the conditions of pericentromeric perturbation and I cannot see how it may reinforce the authors' hypothesis.

#4-7 All the minor points were adequately addressed.

Third decision letter

MS ID#: jcs.264181R2

MS Title: CTCF maintains centromere function and mitotic fidelity

Authors: Erin Walsh; Thomas Laskarzewski; Thomas J Maresca; Andrew D Stephens

Article Type: Short Report

Dear Dr Stephens,

I am happy to tell you that your manuscript has been accepted for publication in Journal of Cell Science, pending standard publication integrity checks.